

# jicbioimage: a tool for automated and reproducible bioimage analysis

Tjelvar S. G. Olsson and Matthew Hartley

Computational Systems Biology, John Innes Centre, Norwich, UK, United Kingdom

## ABSTRACT

There has been steady improvement in methods for capturing bioimages. However analysing these images still remains a challenge. The Python programming language provides a powerful and flexible environment for scientific computation. It has a wide range of supporting libraries for image processing but lacks native support for common bioimage formats, and requires specific code to be written to ensure that suitable audit trails are generated and analyses are reproducible. Here we describe the development of a Python tool that: (1) allows users to quickly view and explore microscopy data; (2) generate reproducible analyses, encoding a complete history of image transformations from raw data to final result; and (3) scale up analyses from initial exploration to high throughput processing pipelines, with a minimal amount of extra effort. The tool, jicbioimage, is open source and freely available online at http://jicbioimage.readthedocs.io.

## INTRODUCTION

Over the years microscopy has benefited from many technological advancements. There is now a wide range of differing types of microscopy including bright-field, differential-interface-contrast (DIC), confocal, scanning electron (SEM) and transmission electron (TEM).

Although there has been impressive improvement in the development of better means of capturing bioimages, the analysis of bioimages still remains a challenge (*Eliceiri et al., 2012*). It is common for researchers to extract qualitative information from an image by simply looking at it. However, this can result in unconscious bias and is often error prone. In particular, our brains have been conditioned to infer patterns (*Witkin & Tenenbaum, 1983*) and our eyes are ill suited to distinguish colour intensities (*Gouras & Zrenner, 1981*), both traits make us bad at quantitative analysis.

For more advanced image analysis, researchers most commonly make use of ImageJ/Fiji (*Schindelin et al., 2012*; *Schneider, Rasband & Eliceiri, 2012*), simply referred to as Fiji from now on. Fiji has an advantage over generic image analysis software in that it can work directly with images from most microscopy instruments. This is achieved by making use of the Open Microscopy Environment's Bio-Formats API (*Linkert et al., 2010*). This, along with a relatively easy to use graphical interface and a

Corresponding author
Tjelvar S. G. Olsson,
tjelvar.olsson@jic.ac.uk

vibrant user community (*Schindelin et al., 2015*), has made Fiji the standard tool in bioimage analysis.

Beyond Fiji, a wide range of tools, libraries and frameworks exist to solve various aspects of the image analysis problem. These include: (1) tools designed to solve very specific analysis problems, such as quantifying plant phenotypes (*Pound et al., 2013*; *Fahlgren et al., 2015*); (2) tools specifically designed for (bio)image analysis (*Möller et al., 2016*) Imaris (www.bitplane.com/imaris/imaris) (*Marée et al., 2016*); (3) analysis pipelining tools, some dedicated specifically to bioimage analysis (*Carpenter et al., 2006*; *Kvilekval et al., 2010*) Icy (http://icy.bioimageanalysis.org/) and some with more general application but specific adaptations to bioimaging problem domains (*Berthold et al., 2007*); (4) general purpose mathematical or statistical analysis tools with packages designed for image analysis (*R Development Core Team, 2016*), MATLAB (Version 7.10.0; The MathWorks Inc., Natick, MA), Mathematica (*Wolfram Research, Inc., 2016*), Python; (5) software libraries or frameworks providing image analysis methods and transforms (*Lowekamp et al., 2013*; *van der Walt et al., 2014*; *Bradski, 2000*). Some of the tools and libraries can be integrated with interactive data exploration tools that aid reproducibility such as Jupyter Notebook (NumFOCUS Foundation, http://jupyter.org), Beaker Notebook (Two Sigma Open Source, http://beakernotebook.com) and Vistrails (https://www.vistrails.org/).

## MOTIVATION

In our work, providing bioimage analysis support to various biology labs, we found that we needed a tool that added functionality to basic Python scripting to:

1. Quickly explore data and test a range of analysis methods step-by-step.
2. Provide easy access to existing powerful image analysis frameworks (such as ITK, scikit-image and OpenCV).
3. Automatically add tracking of image manipulations to enable both auditability and reproducibility.
4. Take small scale (single image or part of image) analyses developed and tested on a desktop/laptop, and transfer them to run on large datasets on a compute cluster or powerful dedicated workstation.

These goals were driven by a desire to be able to quickly explore bioimage data in a reproducible fashion and to be able to easily convert the exploratory work into software for experimental biologists. Reproducibility is key pillar of scientific research and an aspect that is becoming more and more scrutinised as funding bodies strive towards a more open approach.

The available tools all provided some part of this functionality, but non provided all of it, and many of the more complex frameworks required substantial supporting infrastructure to install and run. We needed a lightweight tool, that would enable us to make use of the power of image analysis libraries with Python bindings and Bio-Formats' data conversion capabilities, but without adding substantial installation or maintenance overhead.

We therefore developed a Python tool that enables those with bioimage data to: (1) quickly view and explore their data; (2) generate reproducible analyses, encoding a complete history of image transformations from raw data to final result; and (3) scale up analyses from initial exploration to high throughput processing pipelines, with a minimal amount of extra effort. Here we present our efforts to produce this tool, a Python package named `jicbioimage`.

## METHODS

### Language choice

The `jicbioimage` framework was implemented in Python. Python allows rapid exploration of data and has a rich ecosystem of scientific packages, including several aimed at image analysis; for example scikit-image (*van der Walt et al., 2014*), Mahotas (*Coelho, 2013*), OpenCV (*Bradski, 2000*), and SimpleITK (*Lowekamp et al., 2013*).

### Building on the works of others

Rather than reimplementing existing functionality from scratch, we decided that the framework should form a layer on top of the works of others, see Fig. 1.

Firstly we decided to leverage the Bio-Formats tools for parsing and interpreting bioimage files (*Linkert et al., 2010*). Secondly, we decided that the framework should not directly implement image analysis algorithms. Rather, it should be able to make use of the work done by others in this field. By allowing users of the framework to use the wide range of existing implementations of image analysis algorithms, such as edge detection or thresholding, the framework becomes powerful and flexible.

### Loading bioimage data

Our first goal (allowing rapid exploration of bioimage data) was addressed by providing functionality for loading and manipulating bioimage files. A bioimage file can contain more than one 2D image, often described as 3D, 4D or even 5D data. The framework therefore provides the concept of a microscopy image collection, which contains all the 2D images from the bioimage file. The microscopy image collection can provide information about the number of series, channels, z-stacks and time points in the collection as well as methods for accessing individual images and z-stacks.

This functionality was implemented as a thin wrapper around Bio-Formats' `bfconvert` tool. The `bfconvert` tool can convert many kinds of bioimage file into a set of appropriately named TIFF files. The converted TIFF files are cached in a backend directory. This means that the conversion only needs to happen once, which helps shorten the iteration cycle during the development of new analysis workflows. However, this detail is hidden from the end user who simply needs to write three lines of code to load a bioimage.

```python
from jicbioimage.core.io import DataManager
data_manager = DataManager()
microscopy_collection = data_manager.load("hypocotyl.czi")
```

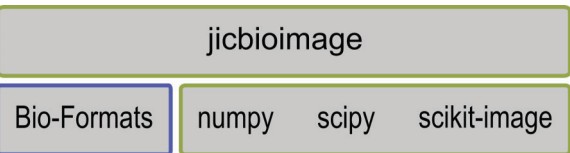

**Figure 1 The `jicbioimage` Python package is a thin wrapper on top of Bio-Formats, `numpy`, `scipy` and `scikit-image`.** It embeds an audit trail for image processing and makes it easy to work with multi-dimensional bioimages.

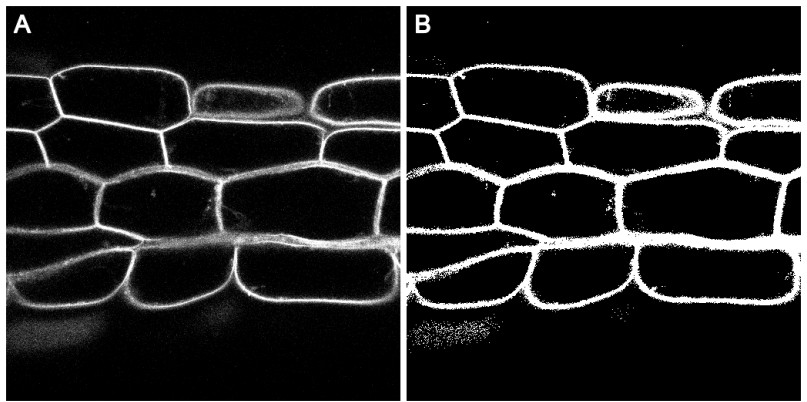

**Figure 2 Section of hypocotyl bioimage.** (A) Cell wall marker signal from channel 1, z-slice 31. (B) Binary image automatically written to disk by the `threshold_abs()` function. Pixels from the input image with an intensity greater than 50 have been set to `True` (white), all other pixels have been set to `False` (black).

The `hypocotyl.czi` file (*Olsson & Calder, 2016*) is a 3D image (contains a number of z-slices) and has information in two color channels. This particular microscopy image is of the hypocotyl of *Arabidopsis thaliana*. The first channel (0) contains the intensity originating from a nuclear marker and the second channel (1) contains intensity originating from a cell wall marker, see Fig. 2A.

Methods for providing access to specific 2D images as well as z-stacks are provided by the microscopy collection.

```
image = microscopy_collection.image(s=0, c=1, z=31, t=0)
zstack = microscopy_collection.zstack(s=0, c=1, t=0)
```

## Reproducible data analysis

To help researchers easily understand how their data are transformed during each step of the data analysis workflow, the framework contains a `transformation` function decorator. This function decorator can be applied to any function that takes an image as input and returns an image as output. When decorating such a function, two pieces of functionality are added to it: (1) the ability to append information about the function itself to the history of the returned image object; and (2) the ability to write the result of the transformation to disk, as a PNG file, with a descriptive name.

Below we illustrate the use of the `transformation` function decorator by implementing a basic threshold transformation and applying it to the image loaded earlier.

```python
from jicbioimage.core.transform import transformation

@transformation
def threshold_abs(image, cutoff):
    """Return thresholded image."""
    return image > cutoff

image = threshold_abs(image, 50)

# Print out the record of how the image was created.
print(image.history.creation)

# Print out each transformation the image has experienced.
for event in image.history:
    print(event)
```

When we apply the `threshold_abs()` transformation, information about this event is appended to the history of the image. Furthermore, the image resulting from the transformation is written to a file named `1_threshold_abs.png` in the working directory, see Fig. 2B.

The history of an image provides a record of how the image was originally created. In this case the image was created by reading in a file representing the 2D image of channel 1, z-slice 31, see the first line in the output below.

```
Created Image from /Users/olssont/projects/jicimagelib_paper/
        scripts/jicbioimage.core_backend/
        a861541517171ae877f08f6dc97bd072/S0_C1_Z31_T0.tif
<History.Event(threshold_abs(image, 50))>
```

The events stored in the history of an image provides an audit log of the transformations that the image has experienced, note the `threshold_abs(image, 50)` event in the script's output above.

Without the additional functionality provided by our transform decorator, we would need to include explicit code to save the image and record the history, for example:

```python
# Import modules needed to create file paths and save images.
import os.path
import skimage.io
```

```
# Define thresholding function that saves the resulting image
# and logs the event.
def threshold_and_log(image, cutoff, output_filename, history):
    history.append("threshold_and_log(image,{})".format(cutoff))
    thresholded_image = image > cutoff
    skimage.io.imsave(output_filename, thresholded_image)
    return thresholded_image, history

# Setup variables to pass to the thresholding function.
history = []
output_filename = "thresholded_image.png"
output_path = "/output"
full_output_filename = os.path.join(output_path, output_filename)

# Call the thresholding function using the image of interest.
image, history = threshold_and_log(image, 50,
        full_output_filename, history)
```

Not only would we need to recreate this code for each step of the image analysis, but we would need to carefully ensure that data paths existed, filenames were consistently constructed and meaningful, and that images were saved in an appropriate format. We would also need to ensure that we recorded that this processing step had taken place and at what stage of processing it had occurred. With the transform decorator, all of these things happen automatically and consistently.

Because the `transformation` function decorator can be applied to any function that has a numpy array as input and output, it is easy to wrap image transforms from other Python libraries. In fact most of the transforms in the `jicbioimage.transform` package are thin wrappers around functions from the scikit-image (*van der Walt et al., 2014*) library.

The numpy array encodes multidimensional numerical data as a block of memory (*van der Walt, Colbert & Varoquaux, 2011*) together with information about the size and shape of that data. Within the Python programming community, it is a widely used standard designed specifically for scientific computation.

The `jicbioimage` framework has an `Image` class that is an extension of this numpy array. One extension is the history property described earlier. Another extension is the addition of a `png()` method, which returns a PNG byte string of the image. This method is used by the `transformation` function decorator to write the transformed image to disk.

The fact that the `Image` class is simply a numpy array makes it easy to work with other scientific Python libraries. In particular libraries such as numpy (*van der Walt, Colbert & Varoquaux, 2011*), scipy (*Jones, Oliphant & Peterson, 2016*; *Oliphant, 2007*; *Millman & Aivazis, 2011*), scikit-image (*van der Walt et al., 2014*), OpenCV (*Bradski, 2000*), simpleITK (*Lowekamp et al., 2013*), and Mohotas (*Coelho, 2013*) that all provide useful algorithms for image analysis.

## Tools for working with segmentations and making annotations

The development of `jicbioimage` has been driven by our need to provide image analysis support across the John Innes Center (www.jic.ac.uk). As such, features have only been added after we have found a repeated use for them. Having added support for working with bioimage data and transformations we found that we continually had to implement custom code for working with segmentations. We therefore implemented a `SegmentedImage` class and a `Region` class. The `SegmentedImage` class inherits from the base `Image` class, but provides its own `png()` function that represents the segmentation as a false colour image. In a `SegmentedImage` a segment is represented by pixels that have the same positive integer. The integer zero is reserved for representing background and all other positive integers represent potential identifiers for segments. The `SegmentedImage` class then provides methods for accessing the set of unique segment identifiers and for accessing the segments as regions by their identifiers.

The `Region` class extends a boolean numpy array representing the mask of the region of interest. It has functionality for accessing a number of useful properties of the region that can be calculated on the fly, for example the area and the perimeter. Some of the properties that can be accessed from a region are themselves instances of the `Region` class, examples include the border and the convex hull.

We found that in communicating with experimental biologists it was useful to be able to create annotated images. We therefore implemented an `AnnotatedImage` class with convenience functions for loading a gray-scale background image, masking out regions of interest, drawing crosses and writing text.

## Technical considerations

In order to facilitate rapid exploration of data and ideas the framework has built-in integration with IPython (*Perez & Granger, 2007*). Our `Image` class (the base 2D unit of processing) provides the methods needed by IPython to enable it to be directly viewable in a IPython/Jupyter notebook (particularly the `_repr_png_()` method). This enables those analysing their image data to see the output of each stage of data analysis and exploration, allowing the framework to act as part of an interactive, reproducible workflow.

The desire to be able to convert exploratory work into tools that can be distributed to experimental biologists was largely satisfied by implementing the framework in Python. Python has extensive support for creating command line tools, graphical user interfaces and web applications.

To aid in reproducibility the framework is under Git version control hosted on GitHub (github.com) and stable releases are available through PyPi (pypi.python.org). This allows scripts developed with the tool to be easily shared and published.

The framework has been designed to be modular. At the top level `jicbioimage` is a name space package into which other sub-packages can be installed. This allows the

dependencies of the sub-packages to be independent of each other. The framework currently has four sub-packages:

- `jicbioimage.core`
- `jicbioimage.transform`
- `jicbioimage.segment`
- `jicbioimage.illustrate`

The framework has been developed using a test-driven approach and has full test coverage. Although this does not mean that the framework is bug free, it gives some level of confidence that existing functionality will not be broken as the framework is developed. The tests are run on Linux and Windows each time the code is pushed to GitHub using the Travis CI (travis-ci.org) and AppVeyor (www.appveyor.com) continuous integration services.

The framework has both high level descriptive as well as API documentation. The documentation is built each time changes are pushed to GitHub using Readthedocs' hosting services Read the Docs (https://readthedocs.org/).

The framework is supported on Linux, Mac and Windows and works with Python 2.7, 3.4, and 3.5. The framework depends on `bftools` and `freeimage` (The FreeImage Project, http://freeimage.sourceforge.net/) as well as the `numpy`, `scipy` and `scikit-image` Python packages.

The code described in this paper was run using:

```
jicbioimage.core==0.13.1
jicbioimage.transform==0.5.1
jicbioimage.segment==0.4.0
jicbioimage.illustrate==0.6.0
numpy==1.10.4
scipy==0.17.0
skimage==0.11.3
```

Detailed installation notes can be found in the online documentation, jicbioimage.readthedocs.io.

## RESULTS

### Usage example

Below is an extended example illustrating some of the aspects of the framework described here. The code segments an image into cells to show the functionality available in the `jicbioimage` Python package.

Segmenting images into cells is a very common problem in bioimage analysis as it enables understanding of cell level properties such as volume and shape. Segmentation is also a requirement if one wishes to locate other features of interest within cells.

First the microscopy data is loaded and the 2D image representing channel 1, z-slice 31 is retrieved.

```python
from jicbioimage.core.io import DataManager

data_manager = DataManager()
microscopy_collection = data_manager.load("hypocotyl.czi")
cellwall = microscopy_collection.image(c=1, z=31)
```

The segmentation protocol will make use of the absolute thresholding transform discussed earlier.

```python
from jicbioimage.core.transform import transformation

@transformation
def threshold_abs(image, cutoff):
    """Return thresholded image."""
    return image > cutoff
```

The segmentation protocol will also make use of a number of transformations built into the `jicbioimage.transform` package. It will also make use of two functions from the `jicbioimage.segment` package. Below is the code for importing these functions.

```python
from jicbioimage.transform import (
    dilate_binary,
    remove_small_objects,
    invert,
)
from jicbioimage.segment import (
    connected_components,
    watershed_with_seeds,
)
```

The code below makes use of the `threshold_abs()` and the imported functions to segment the image.

```python
# Find seeds for the watershed algorithm.
seeds = threshold_abs(cellwall, cutoff=50)
seeds = dilate_binary(seeds)
seeds = invert(seeds)
```

```
seeds = remove_small_objects(seeds, min_size=500)
seeds = connected_components(seeds, background=0)
# Segment the image into cells.
segmentation = watershed_with_seeds(-cellwall, seeds=seeds)
```

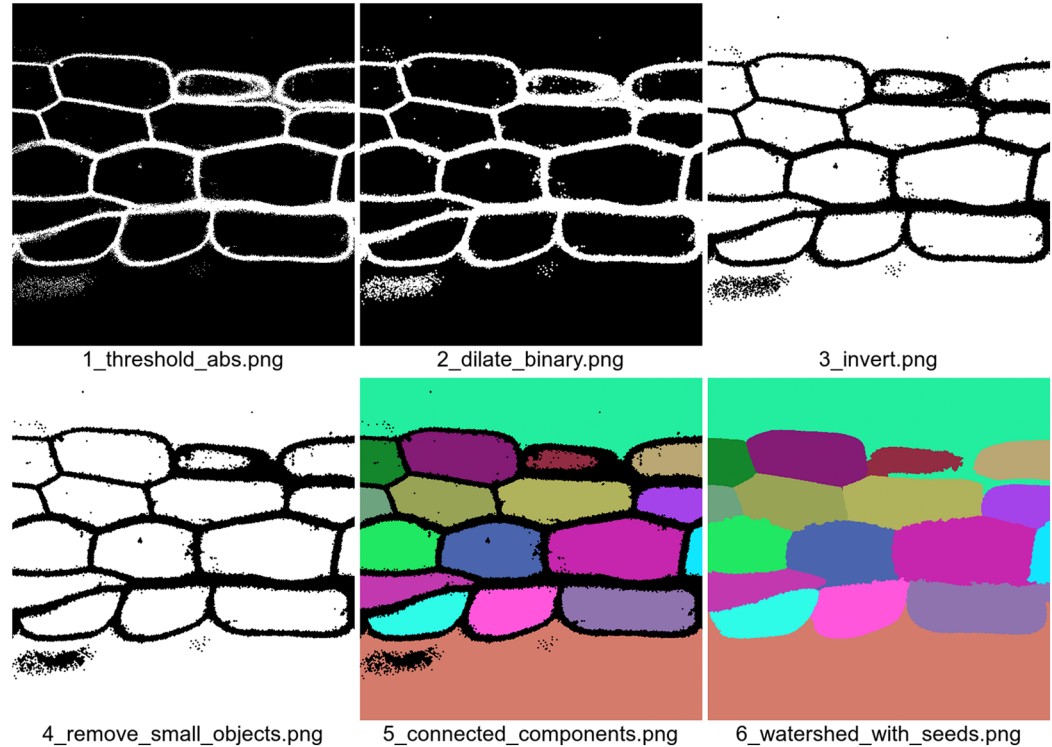

1_threshold_abs.png     2_dilate_binary.png     3_invert.png

4_remove_small_objects.png     5_connected_components.png     6_watershed_with_seeds.png

**Figure 3 Images generated automatically by the segmentation code example.** The filenames, which are also generated automatically, give information about the order in which the images were produced and the specific transformation that was used to produce an image. The fourth transformation removes small white dots in the cell walls.

Because all of the functions imported from the `jicbioimage.transform` and `jicbioimage.segment` packages have been decorated with the `transformation` decorator they automatically write out the resulting images to the working directory, see Fig. 3.

The top and bottom segments correspond to regions that are outside of the hypocotyl tissue. As such we would like to remove them. Here we accomplish this using a simple area filter.

```
# Remove regions from the segmentation that are
# too large to be cells.
for i in segmentation.identifiers:
    region = segmentation.region_by_identifier(i)
```

```
    if region.area > 40000:
        segmentation.remove_region(i)
```

Finally, an augmented image is produced to show the result of the segmentation and the number of pixels of each segmented cell. The code below makes use of the `pretty_color_from_identifier()` function that produces false colour images in a deterministic fashion.

```
from jicbioimage.illustrate import AnnotatedImage
from jicbioimage.core.util.color import pretty_color_from_identifier

annotation = AnnotatedImage.from_grayscale(cellwall)

for i in segmentation.identifiers:
    region = segmentation.region_by_identifier(i)
    outline = region.inner.border.dilate()
    area_str = str(region.area)
    color = pretty_color_from_identifier(i)
    annotation.mask_region(outline, color)
    annotation.text_at(area_str, region.centroid, center=True)

with open("annotation.png", "wb") as fh:
    fh.write(annotation.png())
```

The `png()` method on the image class returns a PNG encoded byte string. One can therefore easily create a PNG file by writing this byte string to a file opened in binary mode. In this case we use it to write out the annotated image, shown in Fig. 4.

In summary this extended example shows that using relatively few lines of code it is possible to:

- load a bioimage and access a particular z-slice from it
- segment the image into cells; automatically creating a visual audit trail
- work with the segmentation to remove large regions that do not represent cells
- create an augmented image displaying the raw data, the segmentation and the size of each cell

## DISCUSSION

Software development inevitably involves a trade-off between implementing functionality directly and including dependent code through libraries or packages. The Python-bioformats (http://pythonhosted.org/python-bioformats/) package exists to provide an interface to OME Bioformats, using the Python Java bridge. We chose not to depend on this package as part of our bioformats integration since managing the Java bridge complicates installation and adds additional code maintenance overhead.
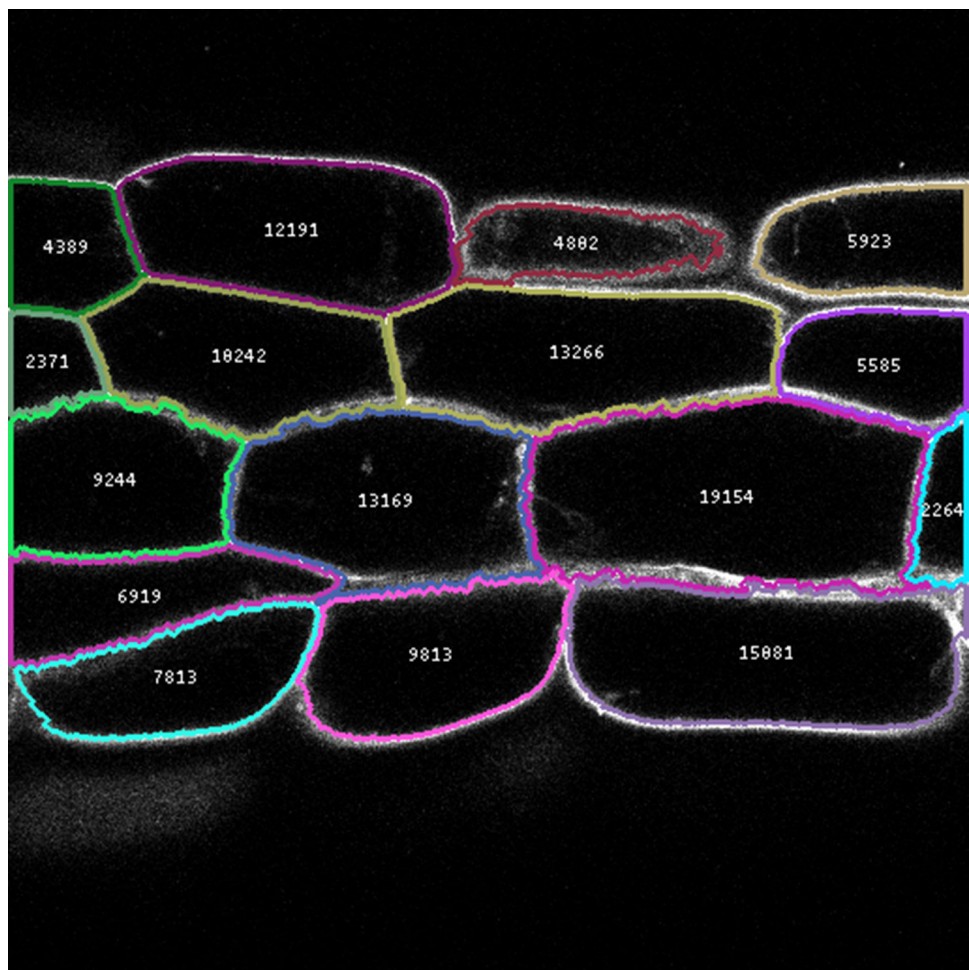

**Figure 4 Annotated image generated by the segmentation code example.** The gray scale intensity represents the cell wall marker signal. Segmented cells are outlined using a deterministic false colouring scheme. The number inside each cell represents the segmented cell area in pixels.

However, as the `jicbioimage` code is relatively modular, it is possible to create another backend. An alternative backend that used the Python-bioformats (http://pythonhosted.org/python-bioformats/) package could therefore be created and added to the `DataManager`. One of the reasons for the code having this modular structure was to be able to add backends when encountering new formats not supported by Bio-Formats. To date we have had to do this once when working with bioimages generated from micro-CT (Computed Tomography) experiments.

We have used our tool to analyse images at a wide range of scales, from the microscopy data presented in this paper (at micrometer scale), though images of whole plant organs (at centimeter scale) to analysis of drone imaging data of whole fields (a scale of multiple meters). This range of scales also cover images captured from varied devices, including different microscopy imagers (confocal microscopes, DIC, SEM/TEM), multispectral cameras, micro-CT capture and drone-mounted cameras. In each case, the support provided by our framework in providing access to image analysis libraries, auditing

and recording image transform history and allowing rapid scaling has been extremely beneficial.

We have used our tool to analyse individual images as well as large collections of bioimages. One recent study required us to analyse over 400 3D images of roots stored in 14 bioimage files ranging in size from 0.5 to 2 GB. In this instance, we prototyped code for segmenting the root into cells and extracting information on a per cell basis on two individual roots. To scale up the analysis we then wrote a script that used the `DataManager` to unpack all the bioimage files and write out a "job list." Using the job list, it was then trivial to parallelize the image analysis.

## Future work

Although we are very happy with our tool, there are still areas where the tool could be improved. These are discussed below.

Use of numpy arrays as a primary data format results in limitations on the scale at which individual datasets can be processed. For example, we have used our tool to analyse micro-CT data. Files produced by the imager can be up to 50 GB in size, at which point the three dimensional structure of the data is too large to fit in memory in a single numpy array. As `jicbioimage` does not load entire zstacks into memory unless specifically asked for, we can work with these large bioimages. However, extending the functionality of our framework to include functionality similar to ImageJ's "virtual stacks" (whereby only part of the whole image resides in memory) would make this process smoother.

At the moment the histories of images are not automatically saved to disk when a script is run. However, since a "history" is essentially a list of strings it is trivial to add a step to save it. We are currently experimenting with ways in which this could be automated. One option is to make use of Python's built in `logging` module and add logging of the history to the `transformation` function decorator. In an ideal world the history would also be stored within the image files saved to disk. This would in theory be possible using the TIFF file format.

At the moment there is no automatic extraction of meta data from bioimages. Although the meta data is always available from the original file it is in many instances useful to have direct access to it from within the data analysis script. In the future, we plan to add functionality to allow this.

## CONCLUSION

The work described in this paper arose out of necessity. As computational scientists supporting a large number of biology groups we needed a tool that would: (1) allow us to quickly view and explore bioimage data; (2) generate reproducible analyses, encoding a complete history of image transformations from raw data to final result; and (3) scale up analyses from initial exploration to high throughput processing pipelines, with a minimal amount of extra effort.

The `jicbioimage` Python package provides all these features as a set of loosely coupled submodules. These allow the user to read in bioimage data, transform and segment

the data whilst having the history recorded both in the image objects and as a series of image files written to disk. Furthermore, the package includes lightweight functionality for creating annotated images to aid in the creation of more informative images.

To date we have used the tool, and its nascent precursors, on over fifteen internal projects at various stages of the publication pipeline (*Berry et al., 2015*; *Duncan et al., 2016*; *Rosa, Duncan & Dean, 2016*). As the package has matured, we have found substantial gains in our productivity. Each feature has resulted in a substantial reduction in the development time for subsequent bioimage analysis projects. We feel that we now have functionality to address most of the bottlenecks in our work. As such we consider the package stable. Furthermore, we consider the package to be a time saving tool that provides provenance and is easy to use.

It is our hope that the tool will be useful to both experimental and computational biologists. The framework developed here is ideally suited to any researcher who desires automation and reproducibility in their bioimage analysis.

## ACKNOWLEDGEMENTS

We thank the JIC Bioimaging facility and staff for their contribution to this publication. We thank Grant Calder for help with the acquisition of the microscopy data. We thank Richard Morris for feedback on the manuscript. We thank the reviewers for their thorough work and their constructive criticism. Their feedback vastly improved the quality of the paper.

### Funding

This research was supported by the Biotechnology and Biological Sciences Research Council. The funders had no role in study design, data collection and analysis, decision to publish, or preparation of the manuscript.

### Grant Disclosures

The following grant information was disclosed by the authors:
Biotechnology and Biological Sciences Research Council.

### Competing Interests

Tjelvar S. G. Olsson and Matthew Hartley are employees of John Innes Centre, Norwich, United Kingdom.

### Author Contributions

- Tjelvar S. G. Olsson conceived and designed the experiments, performed the experiments, analyzed the data, contributed reagents/materials/analysis tools, wrote the paper, prepared figures and/or tables, reviewed drafts of the paper.
- Matthew Hartley conceived and designed the experiments, performed the experiments, analyzed the data, contributed reagents/materials/analysis tools, wrote the paper, reviewed drafts of the paper.

## Data Deposition

`jicbioimage.core`, https://github.com/JIC-CSB/jicbioimage.core;

`jicbioimage.transform`, https://github.com/JIC-CSB/jicbioimage.transform;

`jicbioimage.segment`, https://github.com/JIC-CSB/jicbioimage.segment;

`jicbioimage.illustrate`, https://github.com/JIC-CSB/jicbioimage.illustrate;

*Olsson & Calder (2016)*: hypocotyl3.czi. Figshare.
https://dx.doi.org/10.6084/m9.figshare.3438743.v1.

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
