# Peer review of "jicbioimage: a tool for automated and reproducible bioimage analysis"

_PeerJ, doi:10.7717/peerj.2674_

## Round 0.1 · original submission · Minor Revisions

Please focus in your revision especially on the remarks of both reviewers with respect to your claim that "Furthermore, existing bioimage analysis tools do little to aid researchers in generating audit trails and reproducible analyses." As one of the reviewers points out here are many tools which focus on reproducible analysis. I indeed think that the problems that you claim to have overcome might have been solved already by the software CellProfiler, released in 2005 and re-written from scratch in python in 2011 and cited in 3,000 papers. The comparison with existing software needs to be better elaborated in a revised version of the paper. Please address or comment also on all other issues raised by both reviewers.

·

Basic reporting

Manuscript

The manuscript from Olson and Hartley presents an image analysis library design for bioimages (with a focus on microscopy images). The library, coded in Python, enables an easier processing and tracking of all the transformations made to the images analysed. Although such effort is not the completely novel in the biological sciences (see for instance Cytomine, http://www.cytomine.be/, or PlantCV http://plantcv.danforthcenter.org/), it does provide what seems to be an easy and efficient solution for life scientist working with image data (not restricted to microscopy). The change tracing feature is especially important for reproducibility.

The website presenting the pipeline is well documented and contains all the needed information to install and run the pipeline, which helps a lot.

General comments
* * *
- The language is globally satisfactory, but a careful error-proofing is needed, as typos can still be found throughout the manuscript (see first paragraph for examples).

- I found the introduction to be a little too light. It does not provide a clear picture of what is already existing in the field and why jicbioimage is needed. The only existing tool cited in the paper is FIJI, but other exists (as stated above, Cytomine or PlantCV for examples).

- I would put the link the jicbioimage package already in the abstract, or in a more visible place.

- jicbioimage is used, in the example given in the manuscript, for microscopy data. It might be useful to discuss its use with other type of images (organism scale images for instance)

- I would add a scheme representing how jicbioimage integrate with the existing Python libraries. for non-coders (e.g. end-users) it might not be obvious how the library integrate with these tools. It might also highlight its importances, as a "wrapper" between the existing libraries

- In the same spirit, maybe show an example of the code that would be used without jicbioimage for the same task (for instance for the simple thresholding).


Specific comments
* * *
- l 161: please define or explicit numply array for non-coders.

- I would merge the figures 1 and 2 (1a and 1b)

- The citations style does not correspond to the PeerJ style (Name Year)

Experimental design

No comments

Validity of the findings

No comments

·

Basic reporting

For reasons of time, I did not thoroughly compare against the PeerJ templates. But the article is well written, with good structure and flow.

Experimental design

Software articles are often shoehorned into biological journals whose criteria focus on experimental science. But software engineering is not really experimental science. So many of the criteria in this section end up being rather irrelevant.

That said, the jicbioimage software was clearly crafted to a high technical standard (GitHub, CI, lots of unit tests, etc., built on current powerful technologies), so there is absolutely no issue there.

Validity of the findings

While the jicbioimage project is clearly useful and of high quality, the paper puts forth some incorrect and unsupported claims, which must be addressed before publication.

p4 L8-11
> In particular there is a lack of tools that help researchers create customised bioimage analysis work flows that can be run in an automated fashion. Furthermore, existing bioimage analysis tools do little to aid researchers in generating audit trails and reproducible analyses.

Completely untrue. There are many tools which substantially focus on and support reproducible analysis, such as:

- CellProfiler - http://cellprofiler.org/
- KNIME Image Processing - http://knime.org/ + https://tech.knime.org/community/image-processing
- ImageJ - http://imagej.net/
- Icy - http://icy.bioimageanalysis.org/
- MiToBo - http://www2.informatik.uni-halle.de/agprbio/mitobo/
- Bisque - http://bioimage.ucsb.edu/bisque
- OMERO - https://www.openmicroscopy.org/site/products/omero
- VisTrails - http://www.vistrails.org/index.php/Main_Page
- Jupyter Notebook - http://jupyter.org/
- Beaker Notebook - http://beakernotebook.com/
- R - https://cran.r-project.org/web/views/ReproducibleResearch.html
- MATLAB - http://www.mathworks.com/products/matlab/

The list goes on and on. In particular, many of the tools above (e.g., CellProfiler and KNIME Image Processing) were explicitly designed with reproducibility as a primary goal. To claim that "existing bioimage analysis tools do little to aid researchers in generating ... reproducible analyses" does an incredible disservice to the scientific developer community.

The authors themselves cite [1] (Eliceiri 2012) which reviews several workflow systems and states "the key issues addressed by a workflow system are reproducibility, archivability and the ability to share workflows."

I strongly encourage the authors to drop this claim, and instead focus on what jicbioimage brings to the table: a useful layer on top of the existing power of numpy/scipy/skimage, with embedded audit trails (a nice feature).

p4 L30-32
> In particular, our brains have been conditioned to infer patterns and our eyes are ill suited to distinguish colour intensities, both traits make us bad at quantitative analysis.

While I agree completely, I think a citation is warranted here.

p5 L40-43
> However, there is currently little support [within ImageJ and Fiji] for researchers to create customised bioimage analysis workflows that can be run in an automated fashion. Further more, current bioimage analysis tools do little to aid researchers in generating audit trails and reproducible analyses.

On the contrary, ImageJ offers extensive support for automation/reproducibility. The Script Editor, Macro Recorder and Command Finder make ImageJ very useful for quickly prototyping your image analysis, as well as reproducing it en masse automatically. The Script Editor supports several languages including Jython, a JVM-based flavor of Python (and as an aside: there is even an experimental native Python script language plugin -- https://github.com/scijava/scripting-cpython). Commands written in the ImageJ2/SciJava paradigm automatically work reproducibly across a range of tools in the SciJava ecosystem.

For details, see:

- http://imagej.net/Batch
- http://imagej.net/Scripting_Headless
- http://imagej.net/Script_parameters
- http://imagej.net/Ops
- http://imagej.net/SciJava
- https://imagej.github.io/presentations/2015-09-03-imagej2-and-fiji/

p5 L51-58
> Motivation

While the motivation given is laudible, again, the implication is that there are not already existing tools in the community which achieve these goals, which is not the case. That said, novelty/innovation is explicitly not one of PeerJ's criteria for publication, so this fact is really no problem. My concern here is simply that this paper should put forth an accurate assessment of the technological landscape.

p6 L68-70
> Python is becoming established as the de facto scripting language in scientific computing.

This is a very arguable point, which I recommend cutting. There are lots of good reasons to use Python, and it is very popular. But so are R and MATLAB, as well as several scripting languages on the JVM such as Scala, Groovy and Clojure. And new languages like Julia continue to grow and gain momentum. Technology tends to fragment and diversify rather than converge. It would be best to justify Python for its strengths as the authors do in following sentences, without making explicit claims vs. the many other popular tools. It is enough of a justification that numpy/scipy/skimage are written in Python, and jicbioimage builds on those tools.

p6 L93-94
> The bfconvert tool can convert any bioimage file into a set of appropriately named TIFF files.

While I wish that were true, there will always be some formats which Bio-Formats does not support. It would be best to change "any bioimage file" to "most bioimage files" or perhaps "many kinds of bioimage files."

Additional comments

I have a few comments and questions on the technical design of the software, as well as some other miscellaneous comments.

== Small grammar issues ==

p4
> Over the years microscopy has has benefited
Over the years microscopy has benefited

p5 L37, p6 L77, p6 L92
> BioFormats
Bio-Formats

p8 L122
> transformation to to disk
transformation to disk

== Technical/design issues ==

p6 L92-94
> This functionality was implemented as a thin wrapper around BioFormats’ bfconvert tool. The bfconvert tool can convert any bioimage file into a set of appropriately named TIFF files.

This approach discards the majority of the metadata, particularly bio-specific metadata. Using OME-TIFF instead would mitigate the problem, as would caching the metadata to companion files (e.g., JSON).

I also question why an existing Bio-Formats/Python integration library such as python-bioformats (http://pythonhosted.org/python-bioformats/) was not used. The paper would benefit from a concise statement of tradeoffs between I/O approaches, and why the bfconvert approach was chosen -- e.g. simplicity of implementation and maintenance?

p11 L199-201
> In order to facilitate rapid exploration of data and ideas the framework has built-in integration with IPython [14]. In practise this means that images and collection of images can be viewed directly in the IPython notebook.

What sort of integration? Anything beyond "it's written in Python, so you can call it from a notebook"? Are there specific jicbioimage functions which improve the notebook experience? If not, note that there are many Jupyter kernels these days -- you can also call ImageJ et al from a notebook (https://github.com/4Quant/ImgLib2-Notebooks).

p16 L
> (3) scale up analyses from initial exploration to high throughput processing pipelines, with a minimal amount of extra effort. The jicbioimage Python package provides all these features

This paper offers no discussion or examples of utilizing the jicbioimage framework on high-content / high-throughput / big data, other than indirectly via the single citation [23] (Berry et al 2015). Big data poses many technical challenges, depending on the nature of the data (big planes? lots of planes? many dimensions? etc.). Of course, there may not be room in the paper to expound at length, but it would be helpful to have a short, concrete discussion of the technical capabilities and constraints of jicbioimage in this regard. Limits to numpy array size? Performance issues when scaling up analyses? etc. It would also be very welcome to include a figure illustrating jicbioimage's high throughput capabilities -- e.g., a table of aggregate statistics summarizing a large analysis. Combining figures 1 and 2 could make room for this.

p8 L137-138
> # Print out the record of how the image was created.
> print(image.history.creation)

The embedded audit trail is a really nice feature. It would be helpful to clarify how this history is persisted, since it is crucial metadata about an analysis. The paper makes a point of describing how pixels are saved to disk, but I noticed no mention of if/how metadata is saved. E.g., can you serialize the image objects via pickle? When? Does it scale?

== Other ==

p6 L78-82
> Secondly, we decided that the framework should not directly implement image analysis algorithms. Rather, it should be able to make use of the work done by others in this field. By allowing users of the framework to use the wide range of existing implementations of image analysis algorithms, such as edge detection or thresholding, the framework becomes powerful and flexible.

Very nice to see this understood and emphasized!

p7
> Figure 1

I suggest merging Figure 1 into Figure 2, into a single side-by-side figure.

p16 L351-353
> A possible future development would be a tool to automatically generate schematics of the bioimage analysis workflows (from the scripts), which is something that we still find time consuming.

I encourage the authors to take a very serious look at KNIME (http://knime.org/, https://tech.knime.org/community/image-processing).

Finally, I congratulate the authors on producing a well-written set of software libraries.

---

## Round 0.2 · accepted · Accept

I feel that all remarks and concerns by the reviewers and myself were addressed properly and the paper therefore can be accepted for publication.